# In Vitro Activity of Ebselen and Diphenyl Diselenide Alone and in Combination with Drugs against *Trichophyton mentagrophytes* Strains

**DOI:** 10.3390/pharmaceutics14061158

**Published:** 2022-05-28

**Authors:** Sebastian Gnat, Dominik Łagowski, Mariusz Dyląg, Grzegorz Jóźwiak, Aleksandra Trościańczyk, Aneta Nowakiewicz

**Affiliations:** 1Department of Veterinary Microbiology, Faculty of Veterinary Medicine, University of Life Sciences, Akademicka 12, 20-033 Lublin, Poland; dominik.lagowski@up.lublin.pl (D.Ł.); aleksandra.trosciaczyk@up.lublin.pl (A.T.); aneta.nowakiewicz@up.lublin.pl (A.N.); 2Department of Mycology and Genetics, Faculty of Biological Sciences, University of Wroclaw, S. Przybyszewskiego 63, 50-137 Wroclaw, Poland; mariusz.dylag@uwr.edu.pl; 3Department of Inorganic Chemistry, Faculty of Pharmacy, Medical University of Lublin, Chodźki 4a, 20-093 Lublin, Poland; grzegorzjozwiak@umlub.pl

**Keywords:** dermatophytes, ebselen, diphenyl diselenide, antifungal therapy, *Trichophyton mentagrophytes*

## Abstract

Background: Dermatophytoses are one of the most prevalent infectious diseases in the world for which the pace of developing new drugs has not kept pace with the observed therapeutic problems. Thus, searching for new antifungals with an alternative and novel mechanism of action is necessary. Objective: This study aimed to evaluate the antifungal activity of ebselen and diphenyl diselenide against *Trichophyton mentagrophytes* clinical isolates. Methods: In vitro antifungal susceptibility was assessed for organoselenium compounds used alone or in combination with allylamines and azoles according to the 3rd edition of the CLSI M38 protocol. Results: Ebselen demonstrated high antifungal activity with MIC_GM_ equal to 0.442 μg/mL and 0.518 μg/mL in the case of human and animal origin strains, respectively. The values of MIC_GM_ of diphenyl diselenide were higher: 17.36 μg/mL and 13.45 μg/mL for the human and animal isolates, respectively. Synergistic or additive effects between terbinafine and ebselen or diphenyl diselenide were observed in the case of 12% and 20% strains, respectively. In turn, the combination of itraconazole with diphenyl diselenide showed a synergistic effect only in the case of 6% of the tested strains, whereas no synergism was shown in the combination with ebselen. Conclusions: The results highlight the promising activity of organoselenium compounds against *Trichophyton mentagrophytes.* However, their use in combinational therapy with antifungal drugs seems to be unjustified due to the weak synergistic effect observed.

## 1. Introduction

Superficial fungal infections are one of the most prevalent infectious diseases globally [1]. Literature reports have revealed that dermatomycoses can affect almost one billion people around the world, and approximately $1.67 billion is spent on the treatment of these kinds of infections each year [2,3]. The increased frequency of these diseases recorded in the last years can be attributed to the growing population of patients belonging to so-called risk groups, including pet owners [2,4,5,6]. This phenomenon can be related to a high incidence of asymptomatically infected animals, especially cats, dogs, and guinea pigs [7,8,9]. There are also reports that, once transferred from animal to human, zoophilic dermatophyte infection can be easily transmitted indirectly from human to human [10,11].

Among dermatophytes, *Trichophyton mentagrophytes* stands out as the second most common causative agent of dermatophytosis in humans, after *Trichophyton rubrum* [1]. *T. mentagrophytes* infections in humans are zoonoses in the vast majority of cases [2]. From a taxonomic point of view, *T. mentagrophytes sensu stricto* is a member of the *T. mentagrophytes* complex, which includes seven species: *T. tonsurans*, *T. mentagrophytes*, *T. interdigitale*, *T. equinum*, *T. quinckeanum*, *T. schöenleinii*, and *T. simii* [1,4]. These species differ with regards to their ecological preferences [2]. *Trichophyton mentagrophytes sensu stricto* is a zoophilic species and the second most common causal agent of dermatophytosis from this complex, i.e., *Trichophyton interdigitale* is anthropophilic [1,4,6,8]. Discriminating between these two species is difficult, and often, in addition to morphological features, diagnostics should refer to the sources of infection [4,7]. Moreover, to ensure identification accuracy, the use of molecular methods is recommended in combination with morphological analyses [2,4].

Currently, there are numerous options for the treatment of dermatophyte infections [12,13,14]. In its initial stages, the topical formulations of azoles and/or allylamines are generally sufficient to achieve therapeutic success [15,16]. Terbinafine or naftifine used in combination with azole drugs, i.e., ketoconazole or luliconazole, are regarded as effective and cost-justified strategies to maintain a cured status [15,17]. Other useful options are griseofulvin, amorolfine, and ciclopirox [18]. However, the long duration of treatment, toxicity, and the side effects, in combination with some other drawbacks of conventional therapy, lead to frequent abandonment or complete failure in patients [4,19]. Furthermore, an increase in the number of cases caused by *T. mentagrophytes* resistant to terbinafine or azoles, which are the treatment of choice, as well as a multi-drug resistant phenotype, has been reported [2,4]. Hence, the search for new alternative therapies and antimycotics is of the utmost importance.

Selenium is an essential micronutrient for humans and animals assimilated in inorganic and organic forms with ingested food, as it is present in vegetables, meats, seafood, and nuts [20]. In humans, this microelement is an antioxidant involved in immunological and inflammatory processes and in oxidative stress responses [21]. Moreover, organoselenium compounds are widely studied as they have many potential pharmacological applications due to their antimicrobial activity against several bacterial and fungal pathogens, with promising effects on human cells in terms of therapeutic dosage [20,22,23]. In the literature, it is suggested that the mechanism of action of these organoselenium compounds in fungal cells is related to pro-oxidant activity, which causes intracellular accumulation of reactive oxygen [22,23]. Furthermore, a different mechanism of action has been proposed in the case of yeast cells. It consists in the inhibition of H^+^ ATPase, which consequently changes membrane permeability and leads to cellular death [24]. Contrarily, inorganic selenium compounds are widely used as food supplements, but their use as potential drugs and/or antifungals is limited due to their high toxicity [23]. Ebselen (C_13_H_9_NOSe) and diphenyl diselenide (C_12_H_10_Se_2_) are prospective new antimicrobial drugs from the organoselenium group [24,25,26]. Currently, ebselen is used in the therapy of cardiovascular diseases, arthritis, atherosclerosis, cancer, and bipolar disorder [27]. Furthermore, the in vitro antifungal activity of this compound has also been reported [28,29,30]. In turn, the clinical use of diphenyl diselenide is less well characterized, but its antifungal activity has been demonstrated against *Fusarium* spp., *Candida* spp., *Cryptococcus* spp., *Pythium* spp., *Aspergillus* spp., *Trichosporon* spp., and *Sporothrix* spp. [22,28,31,32]. However, the activity of organic selenium compounds against dermatophytes has been relatively poorly studied [23,26].

Since a growing number of recalcitrant zoophilic dermatophytoses caused by *Trichophyton mentagrophytes* have recently been observed in Europe, this study aimed to determine the in vitro antifungal effectiveness of ebselen and diphenyl diselenide used alone and in combination with terbinafine and itraconazole. In vitro antifungal susceptibility tests were performed according to the Clinical and Laboratory Standards Institute (CLSI) document M38, 3rd edition. The interactions between the antifungals and the organoselenium compounds were evaluated based on a microdilution checkerboard assay following the protocol published in Clinical Microbiology Procedures Handbook, 4th edition.

## 2. Materials and Methods

### 2.1. Materials

Terbinafine (TRB), itraconazole (ITC), ebselen (EBS), and diphenyl diselenide (DPDS) were purchased from Merck Life Sciences (Darmstadt, Germany) and were of analytical grade with at least 99% chemical purity. Microbiological media for the culturing and identification of dermatophytes, i.e., Sabouraud dextrose agar (SDA) and potato dextrose agar (PDA), were purchased from Oxoid (Basingstoke, UK), and malt extract agar (MEA) was purchased from BioMaxima (Lublin, Poland). Microscopic preparations were examined in Olympus BX51 (Tokio, Japan). For light microscopy, the preparations were examined after lactophenol blue and chlorazol black (Merck Life Sciences, Darmstadt, Germany) staining. Calcofluor white (Merck Life Sciences, Darmstadt, Germany) staining was used for fluorescence microscopy. The ITS1 (5′-TCCGTAGGTGAACCTGCGG-3′) and ITS4 (5′-TCCTCCGCTTATTGATATGC-3′) pairs of primers used for the molecular identification tests were synthesized by Genomed (Warsaw, Poland). ITS-PCR was carried out using Qiagen Taq PCR Master Mix (Qiagen, Hilden, Germany) and T Personal Cycler (Biometra GmbH, Goettingen, Germany). The electrophoretic separation of PCR products was carried out in Basica LE agarose obtained from Prona (ABO, Gdańsk, Poland). The ITS sequencing reaction was carried out using a BigDye Terminator Cycle Sequencing Kit (Life Technologies, Carlsbad, CA, USA). The PCR product was purified using an ExTerminator kit (A&A Biotechnology, Gdynia, Poland), and then the DNA sequence was read in a 3500 Genetic Analyser (Life Technologies, Carlsbad, CA, USA). Chemicals for the in vitro antifungal susceptibility tests, including dimethyl sulfoxide, Tween 80, and RPMI, were obtained from Merck Life Science (Darmstadt, Germany). The inoculum supernatants were collected and standardized by counting in a hemocytometer (BrightLine^TM^, Merck Life Sciences, Darmstadt, Germany). The 96-well microtitre plates were purchased from Corning (New York, NY, USA). Minimum inhibitory concentrations (MICs) were read spectrophotometrically using a SmartSpec^TM^ spectrophotometer (BioRad, Hercules, CA, USA).

### 2.2. Dermatophyte Strains

In total, 37 clinical isolates of *Trichophyton mentagrophytes* were obtained from human zoonoses (*n* = 17) and pets (*n* = 20) with typical symptoms of dermatomycosis. All cases of infection were diagnosed in Poland between 2017 and 2020 (Table 1). Human and animal clinical material was collected, especially from the margins of skin lesions, using a sterile surgical scalpel. The dermatophyte isolates showed susceptibility to azoles, including itraconazole, and toward allylamine-type drugs, including terbinafine and naftifine, for which MIC values did not exceed 1 μg/mL.

The identification of these clinical isolates on the species level was performed by a combination of conventional and molecular techniques, comprising the examination of macro- and micro-morphology (Figure 1) and the internal-transcribed spacer (ITS) rDNA region sequencing. First, each sample was examined microscopically after washing in 10% potassium hydroxide (KOH) with dimethyl sulfoxide (DMSO) for the detection of fungal elements. The microscopy slides were viewed under a light microscope (Olympus BX51, Tokyo, Japan) after staining with lactophenol blue and chlorazol black (Merck Life Sciences, Darmstedt, Germany) and under a fluorescence microscope (Olympus BX51, Tokyo, Japan) after staining with calcofluor white (Merck Life Sciences, Darmstadt, Germany) at 300 to 440 nm emission and ca. 355 nm excitation wavelengths. Each preparation was viewed in 10 visual fields at a magnification of 400× and 1000×. The presence of arthrospores or septate hyphae was considered a positive result. The second step included the cultivation and microscopic analysis of pure cultures. Three different microbiological media were used for this purpose, i.e., Sabouraud dextrose agar (SDA), potato dextrose agar (PDA; Oxoid), and malt extract agar (MEA; BioMaxima, Lublin, Poland). The incubations were carried out at 25 °C and 37 °C, and the colonies were analyzed macro- and microscopically every 3 days for 21 days. In the next step, DNA was isolated from the dermatophyte colonies with the phenol-chloroform method. Molecular identification was based on ITS (internal transcribed spacer) sequence analysis. ITS-PCR was carried out in 25 µL of the reaction mixture composed of 12.5 µL Qiagen Taq PCR Master Mix (Qiagen, Hilden, Germany), 10 pmol of each primer, and 100 ng of DNA template. The thermal cycler reaction conditions were as follows: an initial cycle at 95 °C for 3 min followed by 30 cycles at 95 °C for 1 min, 50 °C for 1 min, and 72 °C for 1 min, and then an extension cycle of 72 °C for 10 min. The electrophoretic separation of PCR products was carried out in 1% agarose gels. The PCR product was sequenced using the Sanger method. All the nucleotide sequences obtained were deposited in GenBank (Table 1).

### 2.3. In Vitro Antifungal Susceptibility Tests

In vitro antifungal susceptibility tests were performed according to the Clinical and Laboratory Standards Institute (CLSI) document M38, 3rd edition [33]. The stock solutions of the organoselenium compounds were prepared in dimethyl sulfoxide (DMSO) to reach the final DMSO concentration in the wells below 1%. The final analyzed concentrations were in the range of 0.064–2 μg/mL for ebselen and 2–128 μg/mL for diphenyl disulfide. The preparation of the inoculum suspension containing conidia of *Trichophyton mentagrophytes*, the 2-fold dilutions of the organoselenium compounds, and MIC determination were performed according to CLSI protocol [33] with some modifications. Briefly, the dermatophyte isolates were cultured on Potato Dextrose Agar (Oxoid, Basingstoke, UK) for 21 days, and inoculum suspensions containing only conidia were prepared by gentle scraping mature colonies into sterile physiological saline containing 0.002% Tween 80. Homogeneous inoculum supernatants were collected and standardized by counting in hemocytometer (BrightLine^TM^, Sigma Aldrich, Saint Louis, MI, USA) to achieve final concentration equal to 2 × 10^3^ CFU/mL. The cell suspensions were diluted 1:50 in RPMI 1640 medium and incubated in the presence of indicated concentrations of the organoselenium compounds prepared as serial dilutions within 96-well flat-bottom plates. The last mentioned were incubated at 30 °C for 96 h. The minimum inhibitory concentrations (MICs) were read spectrophotometrically using a SmartSpec^TM^ (BioRad, Hercules, CA, USA) at the 530-nm wavelength (λ). The endpoint for the minimal inhibitory concentration (MIC) was the inhibition of growth corresponding to ≥80% of that of the control free of antifungal compound. *Trichophyton rubrum* MYA4438 and *Trichophyton interdigitale* MYA4439 reference dermatophyte strains were used as quality controls for every new microplate series that was set up. These reference strains were tested against terbinafine (Merck Life Sciences, Darmstadt, Germany). The dilution series of terbinafine was performed simultaneously with the organoselenium compounds using the same laboratory tools.

### 2.4. Interactions between Antifungal Drugs and Organoselenium Compounds

The type of interactions between terbinafine, itraconazole, and the organoselenium compounds were evaluated based on the microdilution checkerboard assay following the standard protocol [34]. Briefly, 2-fold serial dilutions of one of the antifungals were added to rows A–G of a 96-well microtiter plate, whereas dilutions of ebselen or diphenyl selenide were placed in columns 1–9 of the same plate. In this scheme, column 10 and H contained ebselen (or diphenyl diselenide) and an antifungal drug alone, respectively. Columns 11 and 12 were used as positive (without antifungal compound) and negative (without inoculum) controls. It was also confirmed that DMSO at a final concentration equal to 1% did not affect the growth of tested dermatophyte strains that were compared with a positive control. The MIC_80_ values were used for calculation of the Fractional Inhibitory Concentration Index (FICI). The 80% endpoint for the minimal inhibitory concentration (MIC) was read spectrophotometrically using a Varioskan LUX multimode microplate reader (ThermoFisher) at the 530-nm wavelength (λ). The FICI was defined as follows: (MIC_A in combination with B_/MIC_A tested alone_) + (MIC_B in combination with A_/MIC_B tested alone_), where A is the antifungal drug and B is the organoselenium compound. The FICI values were interpreted as follows: FICI ≤ 0.5 corresponds to synergism, 0.5 ˂ FICI ≤ 4 means indifference, and FICI > 4.0: antagonism [34].

### 2.5. Statistical Analysis

The differences between MICs and FICIs were evaluated by the nonparametric Wilcoxon paired *t* test using Statistica 13.1 (Statsoft, Warsaw, Poland). The differences were considered statistically significant at *p* ≤ 0.05.

## 3. Results

The results of the in vitro susceptibility testing of *Trichophyton mentagrophytes* isolates to the antifungals and organoselenium compounds are listed in Table 2. The MIC values of terbinafine and itraconazole against all *T. mentagrophytes* isolates were below 1 μg/mL. The geometric means of the antifungal agents for the human and animal origin isolates were 0.01 μg/mL and 0.019 μg/mL for terbinafine and 0.261 μg/mL and 0.135 μg/mL for itraconazole, respectively. Remarkably, lower MIC_90_ values were obtained for terbinafine than for itraconazole (0.016 μg/mL vs. 0.5 μg/mL and 0.032 μg/mL vs. 0.25 μg/mL for the human and animal isolates, respectively). Moreover, ebselen demonstrated antifungal activity against the *T. mentagrophytes* isolates in both groups, with MIC geometric means of 0.442 μg/mL and 0.518 μg/mL for the human and animal strains, respectively. For diphenyl diselenide, the MIC geometric means were higher: 17.36 μg/mL for the human origin isolates and 13.45 μg/mL for the animal origin strains. The statistical analysis revealed significant differences between the susceptibility (MIC geometric means values) of human vs. animal origin isolates to the antifungal drugs and to the tested organoselenium compounds. The MIC values of ebselen and diphenyl diselenide against *Trichophyton rubrum* MYA4438 and *Trichophyton interdigitale* MYA4439 reference dermatophyte strains were 0.5 μg/mL and 32 μg/mL, respectively.

The results for each drug combination are given in Table 3 and Appendix A. The percentage of strains for which synergism between tested compounds has been demonstrated in case of combination of ebselen or diphenyl diselenide with terbinafine were 10.82% and 18.92%, respectively. In turn, for the combination of itraconazole with diphenyl diselenide, synergism was shown only in the case of 5.4% of tested strains, and no synergism was shown in the combination with ebselen. Antagonism between terbinafine and ebselen was observed in the case of 8.1% strains and between terbinafine and diphenyl diselenide for 13.51% studied strains. In turn, for a combination of itraconazole with ebselen or diphenyl diselenide, antagonism was observed in the case of 81.08% and 18.92% of strains, respectively. The FICI values showed indifference for the combination of ebselen and diphenyl diselenide for all the analyzed isolates. Statistically significant differences were observed for the combination of itraconazole with ebselen.

## 4. Discussion

Inorganic selenium is a trace element with an important role in human and animal nutrition due to its biological activity; however, it is more toxic to mammals than selenium in its organic forms [35]. In this context, several organoselenium compounds have been studied. They have exhibited a diversity of beneficial biological effects and pharmacologic potential for mammalian hosts, such as hepatoprotective, antinociceptive, anti-inflammatory, and antioxidant effects [20,21]. The antimicrobial activity of organoselenium compounds has been described recently, highlighting its potential to be used in antifungal therapy alone or in combination with other antifungals against various fungal species of great importance in the medical mycology field [23,24,25]. Benelli et al. [23] pointed out that the importance of organoselenium compounds can be comparable with that of classic antifungal substances since >95% of fungi were inhibited by these compounds in concentrations ≤64 μg/mL. Furthermore, the broad spectrum of ebselen and diphenyl diselenide antifungal activity included yeasts as well as filamentous and dimorphic fungi [22,26,31,36]. However, studies of this subject are still uncommon, especially in the case of dermatophytes [26,28,30,37,38,39]; additionally, most articles available in the literature provide data obtained from analyses of only a few isolates with one of the molecules. Curiously, a topical selenium-based drug, i.e., shampoo containing 2.5% selenium sulphide, is already commercially used in pharmacies for the treatment of dermatophyte infections and dandruff [39]. This fact suggests the need to undertake even more investigations of the role of these promising compounds in fighting this group of fungi. Our study shows for the first time the in vitro ebselen and diphenyl diselenide activity against *Trichophyton mentagrophytes* with human and animal origin and their combinatory effect with terbinafine and itraconazole.

Our findings proved that *T. mentagrophytes* isolates were inhibited by ebselen and diphenyl diselenide used solely. Moreover, lower MIC values were noted for ebselen, i.e., MIC_GM_ = 0.442 and 0.518 μg/mL for the human and animal origin isolates, respectively, than for diphenyl diselenide, which were (MIC_GM_) 17.36 and 13.45 μg/mL, respectively. This observation is consistent with the previous results. In general, the literature reported lower MIC values of ebselen versus diphenyl diselenide against pathogenic fungi [28,37,40]. The MIC_GM_ values for ebselene and diphenyl diselenide against eukaryotic pathogens, without distinguishing between genera and groups of fungi, were estimated at 3.69 and 15.74 μg/mL, respectively [23]. There are no comparative literature data related to the anti-dermatophyte activity of organoselenium compounds. In one study, Wójtowicz et al. [26] found a very wide MIC range of ebselen in relation to *Microsporum* spp., i.e., from 3.4 to above 500 μg/mL. Concerning other filamentous fungi, ebselen showed promising in vitro antifungal activity. Nevertheless, the MIC_GM_ values turned out to be higher than in our study, i.e., 4.87 and 11.59 μg/mL for *Fusarium* spp. and *Aspergillus* spp., respectively [37,41]. Interestingly, ebselen was also active against *Candida auris*, one of the globally emerging multidrug-resistant yeast-like pathogens, in MIC_GM_ concentrations equal to 3.13 μg/mL [30,42]. Based on an extensive literature review, Benelli et al. [23] showed lower MIC_GM_ values of the antifungal activity of ebselen against yeasts-like pathogens than against filamentous fungi, i.e., in the range of 0.29–3.47 μg/mL and 4.87–11.59 μg/mL, respectively. Moreover, for ebselen the ability to prevent biofilm formation in *Candida* spp. [30] was also shown.

The present report evaluated the antifungal activity of ebselen and diphenyl diselenide and in combination with commonly used antifungal drugs from two different groups, i.e., terbinafine and itraconazole. We observed that the combinations of organoselenium compounds with antifungals against *T. mentagrophytes* isolates exhibited low rates of synergism, and in the case of most of tested isolates was observed indifferent interaction, i.e., fractional inhibitory concentration index (FICI) was between 0.5 and less than or equal to 4. Thus, the combination of drugs does not increase the efficacy of in vitro therapy. In turn, the combination of ebselen and itraconazole demonstrated a predominance of antagonistic (in the case of 81.08% of tested strains) interactions against the tested dermatophytes. These results are not consistent with studies on other fungal groups. Felli Kubiça et al. [28] revealed high rates of synergism at the level of 83.33%–96.67% for use by his research group combinations of antifungal agents (caspofungin, itraconazole, and amphotericin B) and diphenyl diselenide against *Trichosporon asahii* strains, even in a fluconazole-resistant pool of strains. Venturini et al. [40] also reported that combinations of ebselen or diphenyl diselenide with amphotericin B exhibited high rates of synergism, i.e., in the case of over 70% of isolates of *Fusarium* spp. However, a combination of diphenyl diselenide with fluconazole demonstrated a predominance of indifferent (50% tested isolates) and antagonistic (40% studied strains) interactions in respect to *Candida glabrata* strains [36]. Thus, it seems that the interdependencies in the type of interactions between organoselenium compounds combined with antifungal drugs cannot be generalized. They rather show specific properties depending on the group of fungi. The mechanism of action of organoselenium compounds has not been fully described. Several studies have demonstrated that ebselen and diphenyl diselenide are capable of mimicking the activity of glutathione peroxidases, which stimulate a rapid oxidative stress response and catalyze formation of reactive oxygen species [20,43]. Nevertheless, there is no consensus on the antifungal activity of these compounds. Billack et al. [44] and Chan et al. [45] demonstrated that ebselen inhibits the plasma membrane H^+^ ATPase pump (Pma1p) in yeast. As proposed by Azad et al. [46], ebselen increases reactive oxygen species levels in yeast by inhibiting the glutamate dehydrogenase (Gdh3) enzyme involved in glutathione synthesis. However, Thangamani et al. [47] did not agree with these results and demonstrated that ebselen reduces intracellular glutathione (GSH) concentrations leading to the dysregulation of redox homeostasis and that deficiency in glutathione biosynthesis exacerbates this mode of action. Similarly, Felli Kubiça et al. [28] found that diphenyl diselenide can also reduce levels of cellular glutathione (GSH) in yeasts, which is the main non-enzymatic antioxidant, consequently causing cell damage through the action of reactive oxygen species [28]. All the studies cited relate to yeast fungi, primarily *Candida albicans*. There are no similar data in the literature on dermatophytes. Nonetheless, the structure and cell wall composition and their modifications play an important role in the antifungal susceptibility and the development of resistance in yeast and filamentous fungi [48]; therefore, it is likely that the mechanisms of action of organoselenium compounds may also differ between these types of fungi. There is still a large research niche in this regard.

Organoselenium compound-based therapy is a promising alternative when commonly known antifungal drugs are not effective, however, it is only possible when toxicity can be reduced. Nogueira et al. [43] suggest that the relative safety of these compounds in human therapy only in a short-term intake regimen. Furthermore, the beneficial effects and toxicity seem to be directly dose-related. Ebselen present acute lethal potency in laboratory rats and mice when administered by the intraperitoneal route in dose LD_50_ of 400 and 340 μmol/kg, respectively [49]. Diphenyl diselenide showed lower toxicity than ebselen in a rat model in intraperitoneal route dose LD_50_ 210 and 1200 μmol/kg, respectively, and haematological and biochemical parameters indicated no detectable toxicity caused by this substance [47]. The toxic doses are high in relation to the MIC values obtained against dermatophytes. Moreover, diphenyl diselenide does not exhibit acute toxic effects when administrated by the subcutaneous route [50]. Thus, topical administration may be an alternative to the use of this compound in the treatment of dermatophytosis. An open question is whether other selenium compounds will not show toxicity at all while maintaining high antifungal activity.

## 5. Conclusions

In conclusion, we demonstrated that ebselen and diphenyl diselenide exhibit strong in vitro antifungal activity against *T. mentagrophytes* isolates of human and animal origin. However, the combination of these organoselenium compounds with azole and allylamine drugs does not lead to synergistic effect increased activity of known antifungal drugs. In this scenario, it is important to conduct further studies to assess the mechanisms by which the toxicity of these organoselenium compounds can be limited, making them a real alternative to currently known antifungal drugs.

## Figures and Tables

**Figure 1 pharmaceutics-14-01158-f001:**
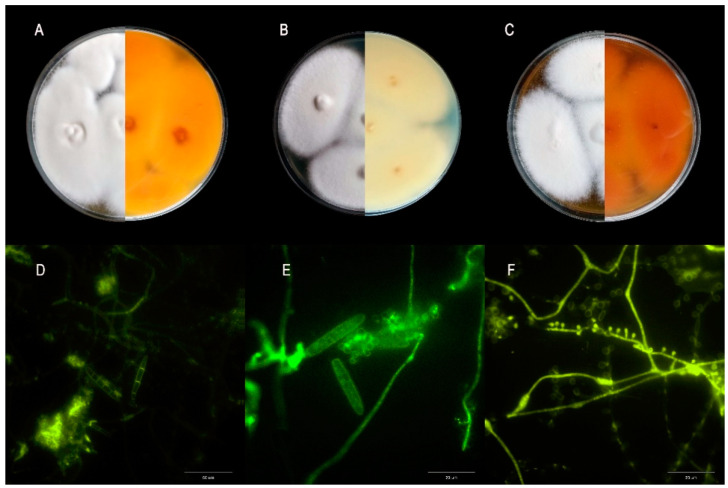
The micro- and macroscopic morphology of *Trichophyton mentagrophytes* after 21 days of incubation. (**A**) Sabouraud dextrose agar (SDA). (**B**) Potato dextrose agar (PDA). (**C**) Malt extract agar (MEA). (**D**–**F**) micromorphology after calcofluor white staining (Olympus BX51, Tokyo, Japan). (**D**) At 400× magnification. (**E**,**F**) At 1000× magnification. Notes: The selection of the three media for the macro- and microscopic identification of dermatophytes was prompted by the characteristic features of the colony and the optimal degree of sporulation. The micromorphological picture of the examined strains shows a distinct transition from the dominant hyphal form in the SDA medium to the sporous form dominating in PDA and MEA. The colonies had a softer powdery or downy texture and a yellow or yellow-orange to brown reverse. There was a visible origin-dependent difference in the colony fluffiness between the strains, which was the highest in the animal isolates. The presence of spherical, one-chambered microconidia, which are more numerous than macroconidia, is an important diagnostic feature. The arrangement lateral to the hyphae is characteristic. In turn, the macroconidia are elongated, cigar-shaped, and multi-chambered.

**Table 1 pharmaceutics-14-01158-t001:** The clinical characteristics of *Trichophyton mentagrophytes* strains of human and animal origin.

Host	Isolates	Accession Number of ITS Sequences	Isolation Source	Sex	Age	Contact with Animals	
Human	TMH1/20	MT106055	tinea capitis	M	71	+	Cat
	TMH3/20	MT106057	tinea capitis	M	68	+	Cat
	TMH4/20	MT106058	tinea capitis	M	20	+	Dog
	TMH7/20	MT106061	tinea unguium	F	65	+	Cat
	TMH8/20	OM574776	tinea capitis	M	19	+	Chinchilla
	TMH9/20	OM574777	tinea corporis	F	43	+	Guinea pig
	TMH10/20	OM574778	tinea corporis	F	37	+	Dog
	TMH1/19	OM574779	tinea capitis	F	36	+	Guinea pig
	TMH3/19	OM574780	tinea corporis	M	21	+	Ferret
	TMH4/19	OM574781	tinea unguium	F	74	+	Chinchilla
	TMH5/19	OM574782	tinea capitis	M	54	+	Cat
	TMH6/19	OM574783	tinea capitis	M	27	+	Ferret
	TMH7/19	OM574784	tinea capitis	F	69	+	Ferret
	TMH10/19	OM574785	tinea capitis	F	64	+	Guinea pig
	TMH11/19	OM574786	tinea capitis	M	22	+	Guinea pig
	TMH12/19	OM574787	tinea capitis	M	26	+	Rabbit
	TMH13/19	OM574788	tinea capitis	F	68	+	Ferret
Guinea pig	TMA13/20	MT106066	torso	M	4	N/A	
	TMA14/20	MT106067	multiple	F	4	N/A	
	TMA15/20	MT106075	multiple	F	6	N/A	
	TMA16/20	MT106076	multiple	F	7	N/A	
	TMA6/19	OM574798	head	M	8	N/A	
	TMA7/19	OM574799	neck	F	5	N/A	
	TMA16/19	OM574800	multiple	M	5	N/A	
Rabbit	TMA1/19	OM574923	head	M	5	N/A	
	TMA28/17	OM574924	head, neck	F	7	N/A	
	TMA18/19	OM574925	head, neck	F	7	N/A	
	TMA19/19	OM574926	multiple	F	3	N/A	
Hamster	TMA21/17	OM574921	abdomen	F	1	N/A	
	TMA31/18	OM574922	torso	M	3	N/A	
Dog	TMA23/17	OM575020	head	M	7	N/A	
	TMA24/17	OM575021	multiple	F	2	N/A	
	TMA12/19	MT106084	neck	F	4	N/A	
	TMA13/19	OM575022	multiple	M	5	N/A	
Cat	TMA25/17	OM574918	torso	M	8	N/A	
	TMA9/19	OM574919	head, neck	F	4	N/A	
	TMA10/19	OM574920	torso	M	7	N/A	

Notes: ITS: internal transcribed spacer; F: female; M: male; +: contact with the animal has occurred; N/A: not applicable.

**Table 2 pharmaceutics-14-01158-t002:** The characteristics of clinical isolates of *Trichophyton mentagrophytes* and the values of the minimum inhibitory concentration (MIC) for known antifungal drugs and organoselenium compounds.

Host	Isolates	Antifungals (µg/mL)						Organoselenium Compounds (µg/mL)					
		TRB			ITC			EBS			DPDS		
		MIC	MIC_50_ MIC_90_ GM	Range Mode	MIC	MIC_50_ MIC_90_ GM	Range Mode	MIC	MIC_50_ MIC_90_ GM	Range Mode	MIC	MIC_50_ MIC_90_ GM	Range Mode
Human	TMH1/20	0.004	0.008 0.016 0.01	0.004–0.032 0.008	0.125	0.25 0.5 0.261	0.125–0.5 0.5	0.25	0.5 1 0.442	0.125–1 0.5	32	32 64 17.36	4–64 32
	TMH3/20	0.008			0.125			0.25			32		
	TMH4/20	0.004			0.25			0.25			8		
	TMH7/20	0.016			0.5			0.5			64		
	TMH8/20	0.016			0.5			0.5			64		
	TMH9/20	0.008			0.125			0.125			16		
	TMH10/20	0.008			0.25			0.5			32		
	TMH1/19	0.008			0.25			0.5			16		
	TMH3/19	0.016			0.064			0.25			8		
	TMH4/19	0.032			0.25			0.5			16		
	TMH5/19	0.016			0.5			1			32		
	TMH6/19	0.004			0.5			1			32		
	TMH7/19	0.016			0.125			1			8		
	TMH10/19	0.008			0.25			0.125			8		
	TMH11/19	0.032			0.5			1			4		
	TMH12/19	0.004			0.5			1			4		
	TMH13/19	0.008			0.5			0.5			32		
Guinea pig	TMA13/20	0.016	0.016 0.032 0.019	0.008–0.064 0.016	0.064	0.125 0.25 0.135	0.032–0.5 0.125	0.25	0.5 1 0.518	0.125–2 1	8	8 32 13.45	4–64 8
	TMA14/20	0.032			0.064			0.125			8		
	TMA15/20	0.064			0.125			0.125			32		
	TMA16/20	0.064			0.25			0.5			32		
	TMA6/19	0.016			0.125			0.5			4		
	TMA7/19	0.008			0.5			1			64		
	TMA16/19	0.064			0.064			1			32		
Rabbit	TMA1/19	0.016			0.125			1			8		
	TMA28/17	0.032			0.25			0.5			8		
	TMA18/19	0.016			0.032			2			16		
	TMA19/19	0.064			0.5			2			64		
Hamster	TMA21/17	0.008			0.125			1			8		
	TMA31/18	0.016			0.125			0.5			16		
Dog	TMA23/17	0.008			0.064			0.5			4		
	TMA24/17	0.008			0.064			1			8		
	TMA12/19	0.016			0.25			0.125			16		
	TMA13/19	0.032			0.5			0.25			32		
Cat	TMA25/17	0.016			0.125			0.25			16		
	TMA9/19	0.008			0.064			1			4		
	TMA10/19	0.008			0.25			0.5			8		

Notes: ITC, itraconazole; TRB, terbinafine; EBS, ebselen, and DPDS, diphenyl diselenide.

**Table 3 pharmaceutics-14-01158-t003:** The fractional inhibitory concentration index (FICI), and the geometric mean (GM) of interactions between antifungal agents and organoselenium compounds against clinical isolates of *Trichophyton mentagrophytes*.

Drug Combination	FICI_GM_	Interaction (%)		
		Synergism	Indifference	Antagonism
TRB + EBS	1.05	10.82	81.08	8.1
TRB + DPDS	1.37	18.92	67.57	13.51
ITC + EBS	8.25 *	0	18.92	81.08
ITC + DPDS	1.91	5.4	75.68	18.92
EBS + DPDS	2.31	0	100	0

Notes: FICI_GM_, geometric mean of fractional inhibitory concentration index; TRB, terbinafine, ITC, itraconazole, EBS, ebselen, DPDS, diphenyl diselenide; and *, significant difference (*p* ≤ 0.05) between groups.

## Data Availability

All data are available from corresponding author upon request.

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
