# Peer review of "In Vitro Activity of Ebselen and Diphenyl Diselenide Alone and in Combination with Drugs against Trichophyton mentagrophytes Strains"

_pharmaceutics, 2022, doi:10.3390/pharmaceutics14061158_

Round 1

Reviewer 1 Report

Congratulations! You did it perfectly and now, I believe that your revision is acceptable for publishing in Pharmaceutics. Please keep in mind to consider mode of action of tested substances against dermatophytes in your future studies. 

Author Response

Reviewer: Congratulations! You did it perfectly and now, I believe that your revision is acceptable for publishing in Pharmaceutics. Please keep in mind to consider mode of action of tested substances against dermatophytes in your future studies.

Authors: Thank you for expressing a good opinion about the article. We have been interested in the topic of drug resistance and alternative forms of therapy in dermatophyte infections for several years. Undertaking research on the mechanisms of action of organoselenium compounds is a very good proposal for future studies.

Reviewer 2 Report

The manuscript, describes “In vitro antifungal activity of ebselen and diphenyl diselenide 2 alone and in combination with commonly applied drugs against 3 human and animal Trichophyton mentagrophytes strains”. The article discusses an in vitro study of two organoselenium compounds when used alone and when combined with two antifungal drugs commonly used in the treatment of Trichophyton mentagrophytes.

The subject of this manuscript is interesting, has novelty and fits the aim and scope of Pharmaceutics Journal. However, the manuscript exhibits several points identified below. If the authors are willing to take into account the comments indicated, they may improve the submitted manuscript:

Title:

The title of the manuscript is so long, if the authors can shorten it, will be better.

The main context:

1-Line 125, in materials and methods, the authors should list in the text the sources and the place where they collect the dermatophyte strains.

2- Table 1, the last column (contact with animals). Please mention that its not applicable for all the animal species not only for (Guinea pig).

3-Table 2, is not well organized. Please reorganize it again in one page to be easy for the reader to follow the results.

4- Line 361, the last paragraph of discussion part the authors mentioned “Organoselenium compound-based therapy is a promising alternative when commonly known antifungal drugs are not effective, however, only when possible toxicity can be reduced”. Its reported that organoselenium compounds behave differently according to their dosage. in this study the concentrations of organoselenium doses used were not mentioned. It’s important to shed light on this point.

5-As the study shown that the organoselenium compounds exhibit strong in vitro antifungal activity against T. mentagrophytes isolates while the combination of these organoselenium compounds with azole and allylamine drugs shows weak synergistic effect. According to these findings, why the authors did not try other classes of antifungal drugs commonly used for dermatophytosis such as griseofulvin and amphotericin B for further investigation of combination therapy with this important organoselenium compound?

Author Response

Reviewer: The manuscript, describes “In vitro antifungal activity of ebselen and diphenyl diselenide 2 alone and in combination with commonly applied drugs against 3 human and animal Trichophyton mentagrophytes strains”. The article discusses an in vitro study of two organoselenium compounds when used alone and when combined with two antifungal drugs commonly used in the treatment of Trichophyton mentagrophytes.

The subject of this manuscript is interesting, has novelty and fits the aim and scope of Pharmaceutics Journal. However, the manuscript exhibits several points identified below. If the authors are willing to take into account the comments indicated, they may improve the submitted manuscript:

Authors: Thank you for expressing a positive opinion about the article

Reviewer: Title: The title of the manuscript is so long, if the authors can shorten it, will be better.

Authors: The title has been shortened

Reviewer: The main context: 1-Line 125, in materials and methods, the authors should list in the text the sources and the place where they collect the dermatophyte strains.

Authors: The material source information has been added. As for the place, we would like to leave that it was only Poland because within the country the locations were varied and very scattered.

Reviewer: 2- Table 1, the last column (contact with animals). Please mention that its not applicable for all the animal species not only for (Guinea pig).

Authors: All columns have been completed. The abbreviation N/A is used and its explanation below the table

Reviewer: 3-Table 2, is not well organized. Please reorganize it again in one page to be easy for the reader to follow the results.

Authors: Despite attempts, it was not possible to limit the table to one page. The results for the 32 isolates are very comprehensive and we want them to remain detailed. We have detected a problem with converting the file to pdf when the table and data move around. We will pay attention to this during the editing process.

Reviewer: 4- Line 361, the last paragraph of discussion part the authors mentioned “Organoselenium compound-based therapy is a promising alternative when commonly known antifungal drugs are not effective, however, only when possible toxicity can be reduced”. Its reported that organoselenium compounds behave differently according to their dosage. in this study the concentrations of organoselenium doses used were not mentioned. It’s important to shed light on this point.

Authors: The last paragraph has been modified as suggested by the reviewer.

Reviewer: 5-As the study shown that the organoselenium compounds exhibit strong in vitro antifungal activity against T. mentagrophytes isolates while the combination of these organoselenium compounds with azole and allylamine drugs shows weak synergistic effect. According to these findings, why the authors did not try other classes of antifungal drugs commonly used for dermatophytosis such as griseofulvin and amphotericin B for further investigation of combination therapy with this important organoselenium compound?

Authors: Currently, allylamines and azoles are used much more frequently in the treatment of dermatophytosis. Terbinafine is even a first-line drug due to its high efficiency. In vitro tests of the drug classes indicated by the Reviewer would certainly be interesting, but for this study we selected the drugs used most often.

Round 2

Reviewer 2 Report

Thanks for taking all the comments in your consideration. The manuscript is improved at its present form.

This manuscript is a resubmission of an earlier submission. The following is a list of the peer review reports and author responses from that submission.

Round 1

Reviewer 1 Report

Evaluation of the antifungal activity of ebselen and diphenyl diselenide against Trichophyton mentagrophytes clinical isolates was introduced in this manuscript. Antifungal susceptibility of organoselenium compounds used alone or in combination with allylamines and azoles were tested in vitro, respectively. Afterward, the authors concluded that the organoselenium compounds exhibited promising activity against Trichophyton mentagrophytes. However, their combinational therapy with antifungal drugs seems to display a weak synergistic effect. Throughout the full manuscript, I think this manuscript is not suitable for publication in the current version due to simple and insufficient experimental design.

Specific comments:

  1. In the introduction part (Line 39-42), why were irrelevant features of dermatophytosis introduced? The background of dermatophytosis caused by Trichophyton mentagrophytes should be emphasized to fit the title.

  1. In the third paragraph of the introduction, the physiological role of selenium should be introduced more concisely to lead directly to the application of ebselen and diphenyl diselenide in antifungal activity. The basis for selection of methods for determination of antifungal activity should be supplemented.

  1. Line 70-73, references should be cited to explain why the strategies of ebselen and biphenyl diselenide were used in combination or alone. the procedure and purpose of evaluating antifungal activity in vitro should be briefly explained.

  1. In the materials and methods part, it is recommended to write the materials and methods separately, on lines 88-89, “In vitro susceptibility tests” should be the title of the next paragraph in which the punctuation format needs to be changed carefully.

  1. Line 81-84, the microscopic and morphology of Trichophyton mentagrophytes in Fig.1 should be explained briefly, what is the basis for choosing these three different media?

  1. In the results part, the antifungal activity of the positive and negative controls should be complemented, is the reason why the activity of organoselenide compounds is lower than antifungals related to their structure or mechanism of antifungal activity? Are there other evaluation methods for further comparison?

  1. In the discussion part, only the literature of difference in antifungal activity of ebselen antifungal with different species and diphenyl diselenide is used solely or in combination was discussed. However, the reasons for the difference in the antifungal activity of the two organoselenium compounds, the mechanism of antifungal activity, the structure-activity relationship, the toxicity evaluation, and in vivo experiments were not involved.

Author Response

Reviewer: Evaluation of the antifungal activity of ebselen and diphenyl diselenide against Trichophyton mentagrophytes clinical isolates was introduced in this manuscript. Antifungal susceptibility of organoselenium compounds used alone or in combination with allylamines and azoles were tested in vitro, respectively. Afterward, the authors concluded that the organoselenium compounds exhibited promising activity against Trichophyton mentagrophytes. However, their combinational therapy with antifungal drugs seems to display a weak synergistic effect. Throughout the full manuscript, I think this manuscript is not suitable for publication in the current version due to simple and insufficient experimental design.

Authors: Both the selection of the topic and the methodology of the research are supported by the current state of the scientific literature. In addition, my team already has a lot of experience in the field outlined in this article, as we have been dealing with dermatophytes for a very long time. To confirm this fact, we have numerous thematically compatible publications.

Reviewer: Specific comments: In the introduction part (Line 39-42), why were irrelevant features of dermatophytosis introduced? The background of dermatophytosis caused by Trichophyton mentagrophytes should be emphasized to fit the title.

Authors: The information provided is important for at least two reasons. First, the argument of the importance of searching for new forms of dermatophytosis therapy is given. Secondly, it highlights the possibility of asymptomatic carriage, very important in the case of zoonoses. Clinical isolates from such cases constitute a large group in these studies. One paragraph of the characteristics of T. mentagrophytes has been added to the introduction

Reviewer: In the third paragraph of the introduction, the physiological role of selenium should be introduced more concisely to lead directly to the application of ebselen and diphenyl diselenide in antifungal activity. The basis for selection of methods for determination of antifungal activity should be supplemented.

Authors: The paragraph was supplemented with a short indication of the mechanisms of action of the tested compounds. However, I do not fully understand the second part of the suggestion. Susceptibility testing is performed according to the CLSI or EUCAST recommendations. This enables a wide range of possibilities for comparison to other results.

Reviewer: Line 70-73, references should be cited to explain why the strategies of ebselen and biphenyl diselenide were used in combination or alone. the procedure and purpose of evaluating antifungal activity in vitro should be briefly explained.

 Authors: The information indicated by the reviewer was added to the aim of the study.

Reviewer: In the materials and methods part, it is recommended to write the materials and methods separately, on lines 88-89, “In vitro susceptibility tests” should be the title of the next paragraph in which the punctuation format needs to be changed carefully.

Authors: The layout of the materials and methods section has been reformatted as suggested.

Reviewer: Line 81-84, the microscopic and morphology of Trichophyton mentagrophytes in Fig.1 should be explained briefly, what is the basis for choosing these three different media?

 Authors: A comprehensive diagnostic description is provided under Fig. 1

Reviewer: In the results part, the antifungal activity of the positive and negative controls should be complemented, is the reason why the activity of organoselenide compounds is lower than antifungals related to their structure or mechanism of antifungal activity? Are there other evaluation methods for further comparison?

 Authors: This is a very helpful point. In the case of reference strains, the results in the CLSI standard are for classic antimycotics, no data for ebselen and diphenyl diselenide are available. Such data can be very useful. The results have been supplemented. Moreover, we did not evaluate the analysis of the mechanisms, so I propose to additionally introduce a paragraph in the discussion, but not in the results section.

Reviewer: In the discussion part, only the literature of difference in antifungal activity of ebselen antifungal with different species and diphenyl diselenide is used solely or in combination was discussed. However, the reasons for the difference in the antifungal activity of the two organoselenium compounds, the mechanism of antifungal activity, the structure-activity relationship, the toxicity evaluation, and in vivo experiments were not involved.

Authors: The discussion has been completed as indicated

Reviewer 2 Report

In this manuscript, Gnat et al. evaluate the antifungal activity of ebselen and diphenyl diselenide used solely or combined with other antifungal drugs, such as terbinafine and itraconazole.

The authors demonstrated the efficiency of ebselen and diphenyl diselenide together with or without commonly applied drugs in dermatophytoses therapy, which can be important for public health. The results are presented clearly, so the article can be publish in the mdpi.

Author Response

Thank you for expressing a good opinion about the article. The topic of searching for the holy grail of therapy against dermatophytes has been of interest to my team for a long time. It seems that scientific research in this area may contribute to solving the important problem of public health, which is currently dermatophytosis.

Reviewer 3 Report

Dear authors,

Please go through attached annotated PDF of you submission, consider corrections in revision and provide point-by-point reply for all comments and suggestions.  Major concerns are:

1- Title should be changed as commented because you have not carried out in vivo study.

2- Detailed methods of conventional and molecular identification of T. mentagrophytes strains should be added.

3- Clinical features of human and animals who were undergone sampling for dermatophytes should be added to revision as a separate Table.

4- The data of combination between ebselen or diphenyl selenide aganist tested T. mentagrophytes strains should be added to revision.

5-  Table 1 should be extended to include detailed data of drug combinations (FICI values) for all 37 T. mentagrophytes strains separately.

6- There is no data about the mode of antifungal action of tested organoselenium compounds in the text. Such data should be added in revision and discuss in relation to different results of susceptibility reported for tested fungi with fungi examined by other researchers as stressed in lines 235 and 246.

Author Response

Reviewer: Dear authors, Please go through attached annotated PDF of you submission, consider corrections in revision and provide point-by-point reply for all comments and suggestions. 

Authors: Thank you for all comments, including editorial comments. We analyzed the entire manuscript and made the necessary corrections. Below are explanations and a description of the adjustments:

Line 54: A detail has been introduced into the sentence

Line 67-68: References added

Paragraph 2.1.: Completely rebuilt. A separate paragraph, Materials has been added

Line 117, 140: A misleading statement has been removed

Line 158: The value that was compared was entered

Table 1: the table was reformatted. Some values skipped between lines due to size

Table 2: Table S1 was added as a complement

Line 235: The entire paragraph up to line 245 is used as a development of this sentence. There seems to be no citation needed here.

Line 246: The discussion was supplemented with the information indicated

References: References have been reviewed and corrections introduced.

Please note that in the revision, due to large changes in the text, the lines may not coincide with the indicated ones. We included lines from the original file in our responses.

Reviewer: Major concerns are: 1- Title should be changed as commented because you have not carried out in vivo study.

Authors: The title was changed as suggested by the reviewer

Reviewer: 2- Detailed methods of conventional and molecular identification of T. mentagrophytes strains should be added.

Authors: The description was supplemented with a detailed description of the individual identification steps. Nevertheless, it seems to us that the description of the PCR mixture and conditions is redundant as it is a very routine method. If the Reviewer thinks otherwise, we can further detail this element.

Reviewer: 3- Clinical features of human and animals who were undergone sampling for dermatophytes should be added to revision as a separate Table.

Authors: This element seems redundant in this article, it adds nothing to the understanding of the results and the discussion. Moreover, in our opinion, the additional table complicates the easy orientation in the results. The present Table 1 lists the host and the type of disease entity. If the Reviewer deems it necessary, we can add such a table in the next stage.

Reviewer: 4- The data of combination between ebselen or diphenyl selenide aganist tested T. mentagrophytes strains should be added to revision.

Authors: The analysis of the combination of ebselen with diphenyl diselenide was not considered to be performed. It seems that due to the toxicity of both compounds, such a combination of substances in one drug is not relevant. This is also confirmed by the analysis of the literature. Rather, organic selenium compounds are an alternative to enhancing the action of classic antimycotics.

Reviewer: 5-  Table 1 should be extended to include detailed data of drug combinations (FICI values) for all 37 T. mentagrophytes strains separately.

Authors: This data is very extensive. We propose to add a detailed table as a supplement, and leave the current version in the article. Additionally, when checking the results, a few calculation errors were corrected, and in Table 2 the degrees of interaction were indicated with an accuracy of two decimal places.

Reviewer; 6- There is no data about the mode of antifungal action of tested organoselenium compounds in the text. Such data should be added in revision and discuss in relation to different results of susceptibility reported for tested fungi with fungi examined by other researchers as stressed in lines 235 and 246.

Authors: Such information is included in the introduction and additionally discussed

Round 2

Reviewer 1 Report

I have no more comments.

Reviewer 3 Report

Manuscript has not been improved properly and the revision is not 

suitable for publishing in Pharmaceutics because it contains

insufficient data about the subject.